biochemistry

tuber mustard, bamboo shoots mustard, baby mustard, glucosinolates, tissue, edible parts

**Authors for correspondence:**
Fen Zhang
e-mail: zhangf_12@163.com
Hao-Ru Tang
e-mail: htang@sicau.edu.cn

This article has been edited by the Royal Society of Chemistry, including the commissioning, peer review process and editorial aspects up to the point of acceptance.
†These authors contributed equally to the study.

# Variations in the glucosinolates of the individual edible parts of three stem mustards (*Brassica juncea*)

Bo Sun[1,†], Yu-Xiao Tian[1,†], Qing Chen[1], Yong Zhang[1], Ya Luo[1], Yan Wang[2], Meng-Yao Li[1], Rong-Gao Gong[1], Xiao-Rong Wang[2], Fen Zhang[1] and Hao-Ru Tang[1,2]

[1]College of Horticulture, and [2]Institute of Pomology and Olericulture, Sichuan Agricultural University, Chengdu 611130, People's Republic of China

BS, 0000-0003-3306-656X; H-RT, 0000-0001-6008-2747

The composition and content of glucosinolates were investigated in the edible parts (petioles, peel and flesh) of tuber mustard, bamboo shoots mustard and baby mustard by high-performance liquid chromatography to reveal the association between the different cooking methods and their glucosinolate profiles. Eight glucosinolates were identified from tuber mustard and baby mustard, including three aliphatic glucosinolates, four indole glucosinolates and one aromatic glucosinolate. Only six of the eight glucosinolates were detected in bamboo shoots mustard. The results show that the distribution and content of glucosinolates varied widely among the different tissues and species. The highest contents of glucosinolates in tuber mustard, bamboo shoots mustard and baby mustard were found in flesh, petioles and peel, respectively. The content of total glucosinolates ranged from 5.21 $\mu$mol g$^{-1}$ dry weight in bamboo shoots mustard flesh to 25.64 $\mu$mol g$^{-1}$ dry weight in baby mustard peel. Aliphatic glucosinolates were predominant in the three stem mustards, followed by indole and aromatic glucosinolates. Sinigrin was the predominant glucosinolate in the three stem mustards. Sinigrin content in tuber mustard was slightly higher than that in baby mustard and much higher than that in bamboo shoots mustard, suggesting that the pungent-tasting stem mustards contained more sinigrin. In addition, a principal components analysis showed that bamboo shoots mustard was distinguishable from the other two stem mustards. A variance analysis indicated that the

glucosinolates were primarily influenced by a species × tissue interaction. The correlations among glucosinolates were also analysed.

# 1. Introduction

The first description of glucosinolates was reported at the beginning of the seventeenth century in mustard seed [1]. Now, these compounds are studied widely as secondary metabolites with complex functions [2]. More than 200 glucosinolates have been identified so far, and most are found in *Brassica* [3]. Glucosinolates contain a sulfated isothiocyanate group, thioglucose and an R-group derived from amino acids [4]. Glucosinolates can be divided into three classes based on the R-group as aliphatic, with a methionine, isoleucine, leucine or valine precursor; indole, with a tryptophan precursor; and aromatic, with a phenylalanine or tyrosine precursor [5]. Myrosinase is a thioglucosidase stored in a different cell, physically separated from its substrate glucosinolates. Glucosinolates that are not hydrolysed are non-toxic, but when plants tissues are damaged by chewing or other actions, the myrosinase is released to hydrolyse the glucosinolates to glucose, sulfate and several toxic/pungent products, including isothiocyanates [6]. Unstable chemical isothiocyanate products have a demonstrated role in carcinogenesis resistance and inhibition [7,8].

Mustard (*Brassica juncea*) is one of the most commonly consumed *Brassica* vegetables with a long planting history in China. Stem mustard is a class of mustard that takes the abnormal stem as the edible part. Tuber mustard (*B. juncea* var. *tumida*), bamboo shoots mustard (*B. juncea* var. *crassicaulis*) and baby mustard (*B. juncea* var. *gemmifera*) are three important variants of stem mustard, and they are typical vegetables in southwest China. The cooking methods of the three stem mustards are generally different due to their diverse flavours. Specifically, tuber mustard is generally fermented and preserved into Szechuan pickles, which are known for their unique and rich flavour. Bamboo shoots mustard is generally eaten fresh due to its light flavour, while those two cooking methods both exist for baby mustard with middle flavour among the three stem mustards. The three stem mustards are rich in glucosinolates like other *Brassica* vegetables, such as cabbage, broccoli, kale and radish [9,10]. The hydrolysis products of glucosinolates lend a strong flavour effect to *Brassica* vegetables [11]. A large number of studies on glucosinolates in *Brassica* vegetables have been reported in recent years [12], and the glucosinolates of tuber mustard have been identified [13]. The variations of glucosinolate content in different tissues were also generally reported [5]. In this study, we first report the glucosinolate composition and contents in bamboo shoots mustard and baby mustard. In addition, the differences in glucosinolates among the three stem mustards and their individual edible parts were compared for the first time. Our results reveal the correlation between glucosinolate content and the cooking methods for tuber mustard, bamboo shoots mustard and baby mustard.

# 2. Experimental set-up

## 2.1. Plant materials

The tuber mustard cultivar (Jinyou), the bamboo shoots mustard cultivar (Chuanyou Bangcai) and the baby mustard cultivar (Linjiang Ercai) were selected as the plant materials in this study. They were grown in an open field at the experimental base located at Sichuan Agricultural University, China, sowed on 2 September 2016 and harvested on 25 December 2016. After harvest, the mustard plants with similar size, in addition without disease and damage, were taken to the laboratory immediately, and the edible parts were retained and manually divided into petioles, peel and flesh, then immediately frozen in liquid nitrogen. Five plants were grouped as a replicate and there were three replicates for each stem mustard. All samples were frozen at −80°C, lyophilized, ground to powder and stored at −20°C.

## 2.2. Sample preparation

The samples were extracted according to the method of Cai *et al.* [14]. Briefly, the freeze-dried samples were boiled for 20 min in 1 ml of distilled water. The supernatant was collected after centrifugation (5 min, 7000$g$), and the precipitate was washed once with 1 ml water and combined with the previous extract after centrifugation. The extract was applied to a DEAE-Sephadex A-25 (40 mg) column (pyridine acetate form) (GE Healthcare, Piscataway, NJ, USA). The glucosinolates were converted into

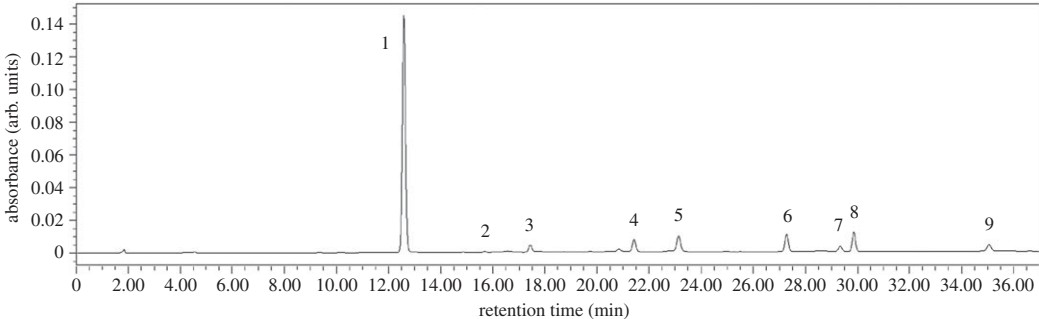

**Figure 1.** High-performance liquid chromatography profile of desulfoglucosinolates in baby mustard peel. 1, Sinigrin; 2, glucoalyssin; 3, gluconapin; 4, 4-hydroxyglucobrassicin; 5, internal standard; 6, glucobrassicin; 7, gluconasturtiin; 8, 4-methoxyglucobrassicin; 9, neoglucobrassicin.

their desulfo analogues by overnight treatment with 100 µl of 0.1% (1.4 units) aryl sulfatase (Sigma, St Louis, MO, USA), and the desulfoglucosinolates were eluted twice with 0.5 ml water.

## 2.3. High-performance liquid chromatography analysis

The HPLC analysis of the desulfoglucosinolates was carried out using a Waters HPLC instrument equipped with a model 2996 PDA absorbance detector (Waters, Milford, MA, USA). The samples (10 µl) were separated at 30°C on a Waters Spherisorb C18 column (250 × 4.6 mm i.d.; 5 µm particle size) using acetonitrile and water at a flow rate of 1.0 ml min$^{-1}$. The procedure employed isocratic elution with 1.5% acetonitrile for the first 5 min; a linear gradient to 20% acetonitrile over the next 15 min, followed by isocratic elution with 20% acetonitrile for the final 10 min. Absorbance was detected at 226 nm. Ortho-nitrophenyl-$\beta$-D-galactopyranoside (Sigma) was used as an internal standard for the HPLC analysis [15].

## 2.4. Statistical analysis

Results are shown as mean $\pm$ standard deviation of three replicates. The statistical analysis was performed using the SPSS v. 18 software program (SPSS Inc., Chicago, IL, USA). Data were analysed using two-way analysis of variance (ANOVA). The least significant difference test was applied to assess differences ($p < 0.05$). A principal components analysis (PCA) and partial least-squares discriminant analysis (PLS-DA) were performed using SIMCA-P 11.5 Demo software (Umetrics, Malmö, Sweden) with unit variance scaling to determine the relationships among the samples. Variance and correlation analyses were also performed.

# 3. Results

## 3.1. Glucosinolate composition in the mustards

The glucosinolates in tuber mustard, bamboo shoots mustard and baby mustard were separated by HPLC. Figure 1 shows the chromatographic peaks for the glucosinolates in baby mustard peel, which is a tissue where all glucosinolates have been found. Eight glucosinolates were detected in tuber mustard and baby mustard, including three aliphatic glucosinolates (sinigrin, glucoalyssin and gluconapin), four indole glucosinolates (4-hydroxyglucobrassicin, glucobrassicin, 4-methoxyglucobrassicin and neoglucobrassicin) and one aromatic glucosinolate (gluconasturtiin). Only six of the eight glucosinolates were detected in bamboo shoots mustard, whereas glucoalyssin and gluconapin were not detected. All identified glucosinolates were typical of *Brassica* vegetables. Sinigrin was the predominant glucosinolate in the three stem mustards, whose content ranging from 59.89% in bamboo shoots mustard peel to 90.14% in baby mustard flesh (electronic supplementary material, figure S1).

## 3.2. Total glucosinolates

The total glucosinolate distribution in the three stem mustards was absolutely different, as shown in figure 2. The results show significant differences in the glucosinolate contents among the different

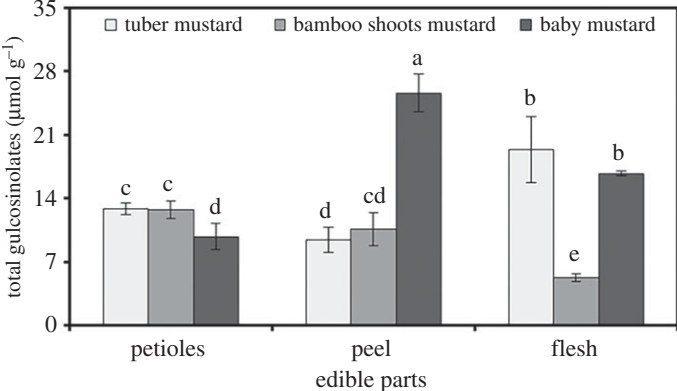

**Figure 2.** Total glucosinolate contents in different tissues of the three stem mustards. Bars with the same letters are not different between samples ($p > 0.05$).

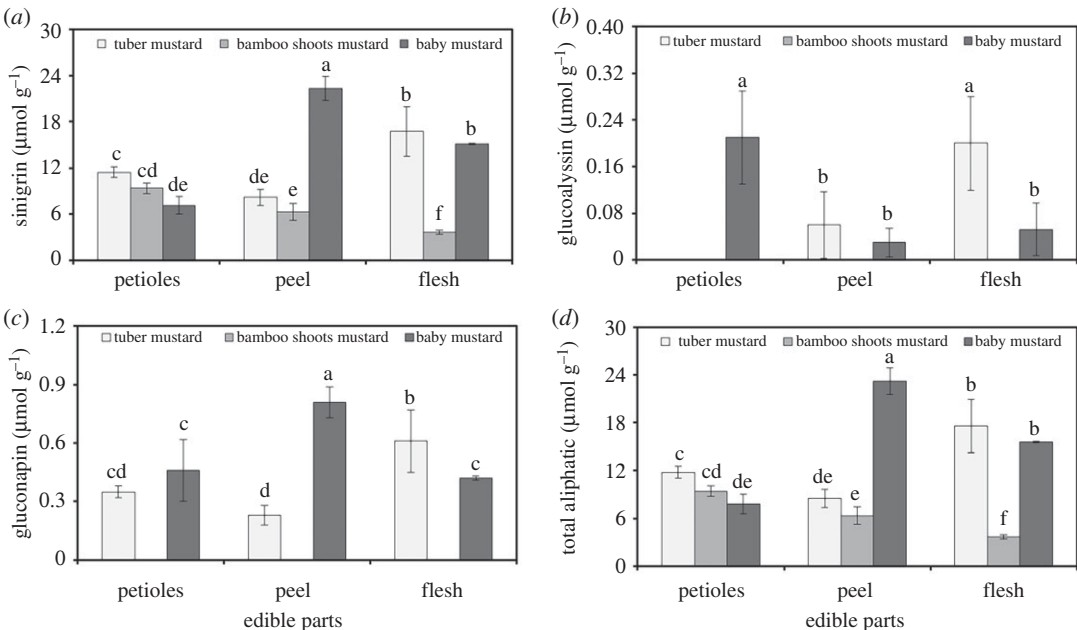

**Figure 3.** The contents and distribution of aliphatic glucosinolates in different tissues of the three stem mustards. (a) Sinigrin, (b) glucoalyssin, (c) gluconapin and (d) total aliphatic glucosinolates. Bars with the same letters are not different between samples ($p > 0.05$).

tissues of tuber mustard and baby mustard. The total glucosinolate content in the flesh of bamboo shoots mustard was remarkably lower than that in the petioles and peel; however, no significant differences were observed between the petiole and peel. The total glucosinolate contents in two of the three stem mustards were significantly different from the other in the same tissue. The total glucosinolate contents in the petioles of tuber mustard and bamboo shoots mustard were significantly higher than those in baby mustard. The total glucosinolate content in the peel of baby mustard was higher than that in tuber mustard and bamboo shoots mustard, and those in the flesh of tuber mustard and baby mustard were higher than that in bamboo shoots mustard. The highest contents of glucosinolates in tuber mustard, bamboo shoots mustard and baby mustard were found in the flesh, petioles and peel, respectively. The total glucosinolate contents ranged from 5.21 $\mu$mol g$^{-1}$ dry weight in the flesh of bamboo shoots mustard to 24.64 $\mu$mol g$^{-1}$ dry weight in the baby mustard peel.

## 3.3. Aliphatic glucosinolates

The aliphatic glucosinolate contents and distribution in different tissues of the three stem mustards are presented in figure 3. The results show that three aliphatic glucosinolates, such as sinigrin, glucoalyssin

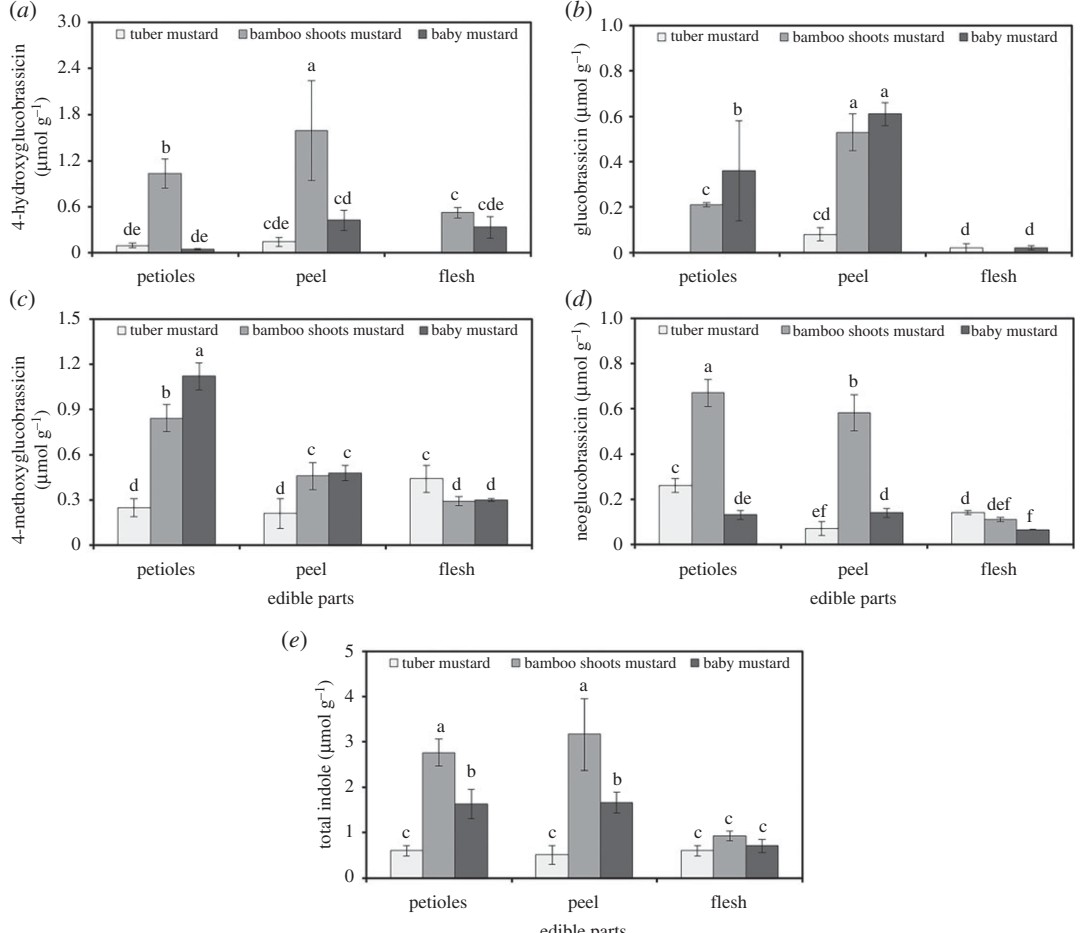

**Figure 4.** The contents and distribution of indole glucosinolates in different tissues of the three stem mustards. (*a*) 4-hydroxyglucobrassicin, (*b*) glucobrassicin, (*c*) 4-methoxyglucobrassicin, (*d*) neoglucobrassisin and (*e*) total indole glucosinolates. Bars with the same letters are not different between samples ($p > 0.05$).

and gluconapin, were detected in tuber mustard and baby mustard. Sinigrin was the major glucosinolate in all tissues, followed by gluconapin. However, only one aliphatic glucosinolate (sinigrin) was identified from bamboo shoots mustard. The distribution of total aliphatic glucosinolates in the three stem mustards was diverse, and a significant difference was observed among the petioles, peel and flesh. The total aliphatic glucosinolate contents ranged from $3.69\ \mu\mathrm{mol\ g}^{-1}$ dry weight in the flesh of bamboo shoots mustard to $23.20\ \mu\mathrm{mol\ g}^{-1}$ dry weight in the peel of baby mustard. In addition, the contents of three individual and total aliphatic glucosinolates in flesh were all significantly higher than those in the petioles and peel of tuber mustard, whereas the contents in peel were all significantly higher than those in the petioles and flesh of baby mustard except for glucoalyssin.

## 3.4. Indole glucosinolates

The content of total indole glucosinolates in bamboo shoots mustard was significantly higher than that in tuber mustard and baby mustard (figure 4). The three stem mustards all contained four indole glucosinolates: 4-hydroxyglucobrassicin, glucobrassicin, 4-methoxyglucobrassicin and neoglucobrassisin. However, the distribution of total indole glucosinolates in petioles, peel and flesh of the three stem mustards was not the same. The total indole glucosinolate contents in the petioles and peel of bamboo shoots mustard and baby mustard were significantly higher than those in flesh, while there were no significant differences among petioles, peel and flesh of tuber mustard. The predominant indole glucosinolate in tuber mustard and baby mustard was 4-methoxyglucobrassicin, whereas that in bamboo shoots mustard was 4-hydroxyglucobrassicin. The total indole glucosinolate content of bamboo shoots mustard peel was the highest ($3.17\ \mu\mathrm{mol\ g}^{-1}$ dry weight), and that of the tuber mustard peel was the lowest ($0.51\ \mu\mathrm{mol\ g}^{-1}$ dry weight).

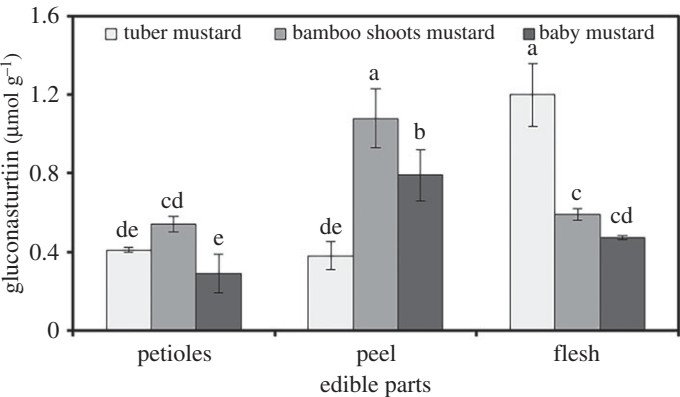

**Figure 5.** The contents and distribution of aromatic glucosinolates in different tissues of the three stem mustards. Bars with the same letters are not different between samples ( $p > 0.05$ ).

## 3.5. Aromatic glucosinolates

Only one type of aromatic glucosinolate, gluconasturtiin, was detected in the tissues of the three stem mustards (figure 5). The distribution of total aromatic glucosinolates was slightly different among the different tissues of the three stem mustards. The total aromatic glucosinolate content was highest in peel, followed by flesh and petioles of bamboo shoots mustard and baby mustard. The total aromatic glucosinolate content in the flesh of tuber mustard was higher than that in the petioles or peel. The total aromatic glucosinolate content ranged from 0.29 μmol g$^{-1}$ dry weight in baby mustard petioles to 1.20 μmol g$^{-1}$ dry weight in tuber mustard flesh.

## 3.6. Principal components analysis and partial least-squares discriminant analysis

PCA is an unsupervised multivariate analysis method performed to determine the effect of species and tissue on glucosinolates (figure 6*a*). The first component (PC1) explained 37.7% of the variance, and the second component (PC2) explained 31.1%. It was possible to discriminate between two groups along PC1 of the score plot: one group consisted of bamboo shoots mustard and the other group consisted of tuber mustard and baby mustard. The petioles and peel of bamboo shoots mustard were divided with the flesh along PC2 of the score plot. However, tuber mustard and baby mustard could not be divided along either PC1 or along PC2 because of considerable overlap.

A further PLS-DA model analysis, which was supervised by species, was subsequently performed. The three groups of all samples were clearly separated on the score plot (figure 6*b*). The three groups consisted of the three stem mustards, and the distribution trend was not similar to the PCA. PLS-DA1 explained 36.4% of variance and PLS-DA2 explained 20.9%. Bamboo shoots mustard was distinguished from tuber mustard and baby mustard by PLS-DA1, while tuber mustard was distinguished from baby mustard by PLS-DA2. The PLS-DA loading plot shows the different contributions of individual glucosinolates to the classification of the three stem mustards (figure 6*c*). According to the results of the loading and VIP values of PLS-DA, the major contributors to bamboo shoots mustard were 4-hydroxyglucobrassicin, neoglucobrassicin and total indole glucosinolates, while gluconapin, 4-methoxyglucobrassicin and glucobrassicin made a greater contribution to baby mustard.

## 3.7. Variance analysis

The variance analysis showed that the species, tissue and their interactive effects played important roles in glucosinolate accumulation, and had significant effects on glucosinolate contents, except for the tissue effects on glucoslyssin and gluconapin (table 1). The tissue effects on glucobrassicin, 4-methoxyglucobrassicin, gluconasturtiin and total aromatic glucosinolates were higher than those of species; however, the proportions of these glucosinolates to total glucosinolates were relatively low. The other glucosinolates were more affected by the species factor compared with the tissue factor. In addition, sinigrin, glucoslyssin, 4-methoxyglucobrassicin, gluconasturtiin, total aliphatic glucosinolates, total aromatic glucosinolates and total glucosinolates were more affected by the species and tissue

**Table 1.** Estimated proportions of the variance components for the glucosinolates in the three stem mustards. Significant differences are marked with asterisks.

| | sinigrin | glucoalyssin | gluconapin | 4-hydroxy glucobrassicin | glucobrassicin | 4-methoxy glucobrassicin | neoglucobrassicin | gluconasturtiin | total aliphatic | total indole | total aromatic | total glucosinolates |
|---|---|---|---|---|---|---|---|---|---|---|---|---|
| species | 0.378** | 0.223** | 0.706** | 0.618** | 0.276** | 0.225** | 0.500** | 0.089** | 0.404** | 0.519** | 0.089** | 0.294** |
| tissue | 0.052** | 0.059 | 0.016 | 0.117** | 0.468** | 0.366** | 0.228** | 0.263** | 0.050** | 0.226** | 0.263** | 0.056** |
| species × tissue | 0.530** | 0.462** | 0.220** | 0.137** | 0.177** | 0.367** | 0.254** | 0.584** | 0.506** | 0.178** | 0.584** | 0.596** |
| deviation | 0.040* | 0.257** | 0.058 | 0.127** | 0.079* | 0.043 | 0.018 | 0.064** | 0.041** | 0.077** | 0.064** | 0.055** |

$*p < 0.05$, $**p < 0.01$.

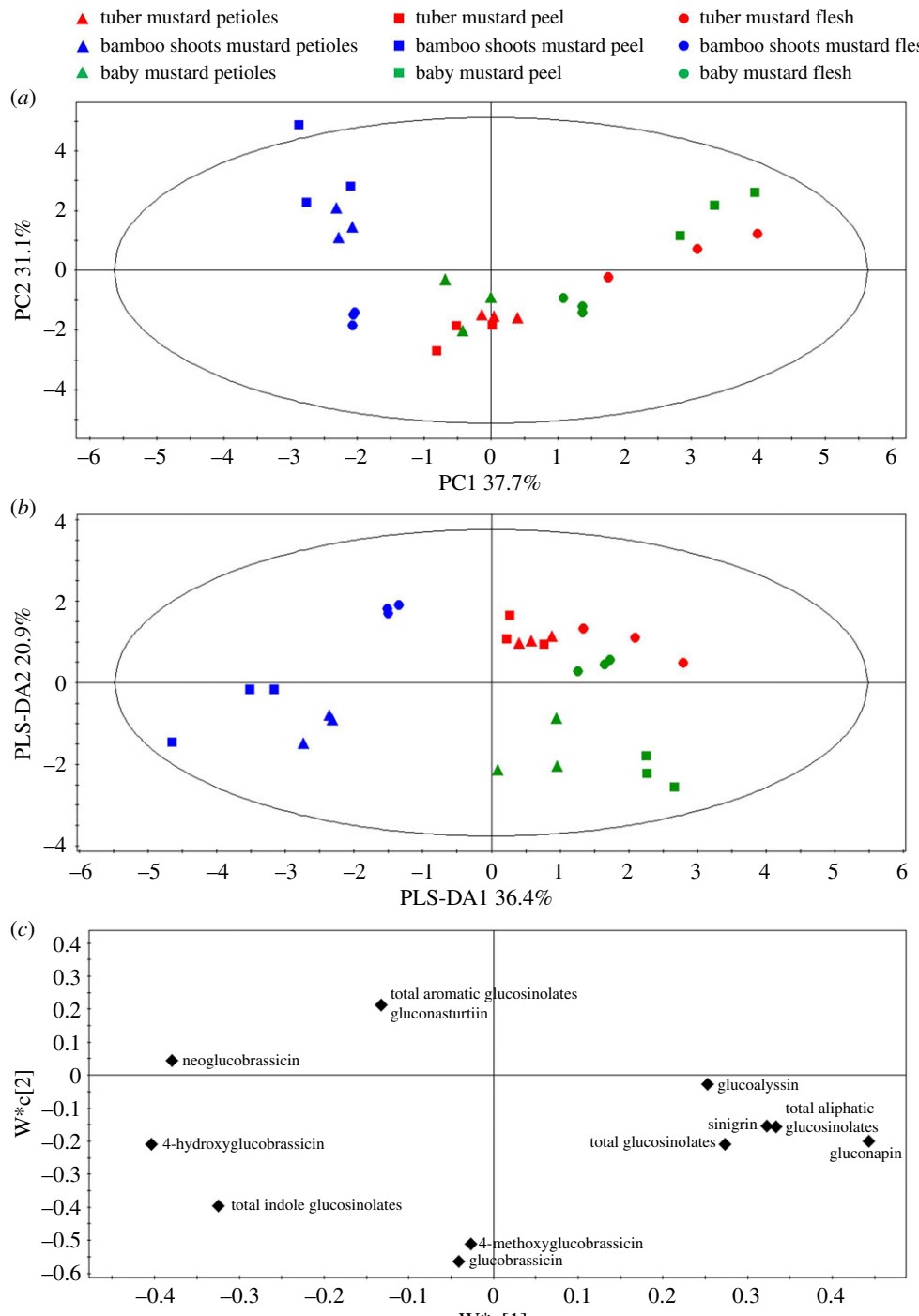

**Figure 6.** PCA and PLS-DA in different tissues of the three stem mustards. (*a*) PCA score plot, (*b*) PLS-DA score plot and (*c*) PLS-DA loading plot. When the position of a substance in loading plot is corresponding to a sample in score plot, the substance has a high content in the sample.

interaction than by a single factor, which almost accounted for 50%. These results indicate that most of the major glucosinolates were affected by the species × tissue interaction.

## 3.8. Correlation analysis

A correlation analysis was performed to investigate the correlations among the different glucosinolates (table 2). A significant positive correlation was detected among the glucosinolates; no significant negative correlation occurred. The results show that the glucosinolate sinigrin contributed greatly to

**Table 2.** Correlation coefficients among tissues with glucosinolates in the three stem mustards.

| glucosinolates | sinigrin | glucoalyssin | gluconapin | 4-hydroxyglucobrassicin | glucobrassicin | 4-methoxyglucobrassicin | neoglucobrassicin | gluconasturtiin | total aliphatic | total indole | total aromatic |
|---|---|---|---|---|---|---|---|---|---|---|---|
| glucoalyssin | 0.159 | | | | | | | | | | |
| gluconapin | 0.857** | 0.518 | | | | | | | | | |
| 4-hydroxy glucobrassicin | −0.316 | −0.579 | −0.627 | | | | | | | | |
| glucobrassicin | 0.212 | −0.05 | 0.203 | 0.492 | | | | | | | |
| 4-methoxy glucobrassicin | −0.131 | 0.472 | 0.042 | 0.09 | 0.438 | | | | | | |
| neoglucobrassicin | −0.267 | −0.425 | −0.572 | 0.819** | 0.346 | 0.327 | | | | | |
| gluconasturtiin | 0.31 | 0.09 | 0.127 | 0.377 | 0.272 | −0.165 | 0.261 | | | | |
| total aliphatic | 0.999** | 0.19 | 0.874** | −0.34 | 0.212 | −0.116 | −0.289 | 0.303 | | | |
| total indole | −0.219 | −0.28 | −0.407 | 0.873** | 0.719* | 0.531 | 0.853** | 0.279 | −0.233 | | |
| total aromatic | 0.31 | 0.09 | 0.127 | 0.377 | 0.272 | −0.165 | 0.261 | 1** | 0.303 | 0.279 | |
| total glucosinolates | 0.985** | 0.151 | 0.819** | −0.182 | 0.343 | −0.04 | −0.139 | 0.402 | 0.983** | −0.058 | 0.402 |

* and ** indicate significance at 0.05 and 0.01 probability levels, respectively.

pungent flavour, had a significant positive correlation with gluconapin, the other contributor to pungent flavour, and both were positively correlated with total aliphatic glucosinolates ($p < 0.01$). Three indole glucosinolates, 4-hydroxyglucobrassicin, glucobrassicin and neoglucobrassicin, were significantly and positively correlated with total indole glucosinolates. The correlation coefficient of gluconasturtiin with total aromatics was 1, as only one type of aromatic glucosinolate was detected in our study. In addition, total glucosinolates were significantly correlated with three glucosinolates (sinigrin, gluconapin and total aliphatic glucosinolates). The correlation coefficient between sinigrin and total aliphatic glucosinolates was the highest (0.999), except for the correlation coefficient between gluconasturtiin and total aromatic glucosinolates.

# 4. Discussion

Significant differences in glucosinolate composition were observed among the different plant species. As reported previously, cauliflower [16], cabbage [17] and Chinese kale [18] have 10, 11 and 13 glucosinolates, and seven, seven and eight of them overlapped with the glucosinolates identified in our study, respectively. The Korean leaf mustard (*B. juncea* var. *integrifolia*) has eight glucosinolates, and four of them were consistent with our results [4]. Furthermore, nine glucosinolates were identified by Li *et al*. [13] from tuber mustard; however, two were different from those detected in our study. We speculate that the composition of glucosinolates is diverse among different plant varieties. The material used in our study was 'Jinyou', whereas four distinct varieties were used in Li *et al*.

As reported by many studies, there were obvious differences in the contents of metabolites among different tissues and species [19–22]. The content of total glucosinolates in fruits of *Raphanus raphanistrum* is 4602 mg $100 \text{ g}^{-1}$ fresh weight, while in leaves, it is only 411 mg $100 \text{ g}^{-1}$ [23]. Our study also shows the diversity of the glucosinolate contents in the three stem mustards among different tissues. The total glucosinolate content was highest in flesh, followed by petioles and peel of tuber mustard, and bamboo shoots mustard contrasted with this distribution. The total glucosinolate content was highest in the peel, followed by flesh and petioles of baby mustard. In addition, the predominant glucosinolate and the proportion of the three groups of glucosinolates differed among species and varieties. The predominant glucosinolate in the three stem mustards was sinigrin; however, the predominant glucosinolate is gluconapin in Chinese kale [18] and most varieties of *Brassica rapa* [24]. That may be because they have different genetic backgrounds, while the composition of aliphatic glucosinolates was mainly affected by genetic factors [25]. Most of the edible parts of *Brassica* vegetables have a high content of aliphatic glucosinolates followed by indole and aromatic glucosinolates [5]. In our study, the proportion of aliphatic, indole and aromatic glucosinolates varied extensively in the three stem mustards. As calculated, the average proportions of total aliphatic glucosinolates in tuber mustard, bamboo shoots mustard and baby mustard were 91.12%, 68.26% and 87.92%, respectively. The total indole glucosinolates accounted for 4.40%, 23.18% and 9.16% of total glucosinolates in tuber mustard, bamboo shoots mustard and baby mustard. The total aromatic glucosinolates accounted for 4.48%, 8.60% and 2.95%, respectively. Interestingly, the content of indole glucosinolates in bamboo shoots mustard was much higher than that in tuber mustard and baby mustard.

Indole glucosinolates have positive defensive effects against insects and microorganisms [26,27]. Previous studies have reported that clubroot has a negative relationship with indole glucosinolates [28], and the morbidities of clubroot in bamboo shoots mustard and baby mustard were 60% and 90%, respectively [29]. In this study, the content of indole glucosinolates in bamboo shoots mustard was higher than that in tuber mustard and baby mustard. Therefore, we inferred the content of indole glucosinolates in the roots of bamboo shoots mustard was higher than that in baby mustard, according to the proportions of indole glucosinolates in petioles, peel and flesh in our study. However, the occurrence of vegetable diseases had a close relationship with variety and cultivation; therefore, this result needs further confirmation.

Among the three stem mustards, the pungent flavour of tuber mustard was the strongest, followed by baby mustard and bamboo shoots mustard. Tuber mustard is often used to ferment pickles, and the pungent glucosinolates are hydrolysed during this process. Baby mustard is not only pickled within 2–3 days, but can also be eaten fresh. Bamboo shoots mustard is customarily eaten fresh or cooked in soup. This eating custom developed according to the characteristic flavour of this mustard. Sinigrin was the predominant glucosinolate in the edible parts of the three stem mustards. Allyl isothiocyanate, the hydrolytic product of sinigrin, plays an important role in the pungent flavour of *Brassica* vegetables [30,31], indicating that sinigrin content has a decisive effect on pungent flavour

[32]. Our results show that sinigrin accounted for 87.79%, 68.26% and 83.55% of total glucosinolates in tuber mustard, bamboo shoots mustard and baby mustard, respectively, confirming that tuber mustard requires the most processing, baby mustard requires less processing and bamboo shoots mustard should be eaten fresh. Allyl isothiocyanate has been reported to be an effective glucosinolate against carcinogenesis [33]. Therefore, we advise consumers to eat some mustard during daily diet. No significant difference was observed between the sinigrin contents in tuber mustard and baby mustard; however, the tuber mustard tastes much more pungent than baby mustard. This may have been due to the high sugar content in baby mustard, which masked the pungent flavour, and tuber mustard perhaps has more amount and activity of myrosinase than those in baby mustard [13,34].

## 5. Conclusion

In the present study, we detected eight glucosinolates from the three stem mustards, and bamboo shoots mustard only contained six. It was the first time that the composition and content of glucosinolates in bamboo shoots mustard and baby mustard have been identified, although glucosinolates in swollen stems of tuber mustard have been reported. Moreover, we compared the glucosinolates in individual edible parts of tuber mustard, bamboo shoots mustard and baby mustard for the first time. The results show that the distribution of glucosinolates varied widely among the different species and tissues. Among all tested samples, the total glucosinolate content ranged from 5.21 $\mu$mol g$^{-1}$ dry weight in bamboo shoots mustard flesh to 25.64 $\mu$mol g$^{-1}$ dry weight in baby mustard peel. Tuber mustard and baby mustard presented with a higher content of aliphatic glucosinolates and a lower content of indole glucosinolates than bamboo shoots mustard. Sinigrin was an important and predominant glucosinolate in the three stem mustards. This study revealed the relationships among different cooking methods and the glucosinolate profiles of the three stem mustards.

Data accessibility. All data used in this manuscript are present in the manuscript and its electronic supplementary material.
Authors' contributions. H.-R.T., F.Z. and B.S. conceived and designed the experiments; Y.-X.T., Q.C., Y.Z., Y.L. and M.-Y.L. performed the experiments; B.S., R.-G.G., Y.W. and X.-R.W. analysed the data; B.S. and Y.-X.T. wrote the paper. All authors gave final approval for publication.
Competing interests. We declare we have no competing interests.
Funding. This work was supported by National Natural Science Foundation of China (31500247).

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
