## [Reviewer comments · Royal Society Open Science]

Review History

RSOS-182054.R0 (Original submission)

Review form: Reviewer 1

Is the manuscript scientifically sound in its present form?

Yes

Are the interpretations and conclusions justified by the results?

Yes

Is the language acceptable?

Yes

Is it clear how to access all supporting data?

Yes

Do you have any ethical concerns with this paper?

No

Have you any concerns about statistical analyses in this paper?

No

Recommendation?

Accept with minor revision (please list in comments)

Comments to the Author(s)

Comments to the Author(s)

This is an acceptable paper for publication in Royal Society Open Science. The manuscript entitled " Variations in the glucosinolates of the individual edible parts of three stem mustards " is an interesting and nice work. The manuscript reported the composition and content of glucosinolates among the edible parts (petioles, peel, and flesh) of tuber mustard, bamboo shoots mustard, and baby mustard. The results also well explained the relationships among the preparation methods and the glucosinolate profiles of the three stem mustards, and this work will provide a reference for the human daily diet.

However, some comments should be made to improve the manuscript. Below are some questions and/or suggestions.

- 1) Line 18-19 on Page 2, as I know it is more than 200 kinds, please check it.
- 2) Line 30-43 on Page 7, what are the genetic relationships between the three mustards in this manuscript and the species you are comparing? What caused different glucosinolate components among them?
- 3) Line 19-32 on Page 8, this is a very interesting phenomenon. The kinds of aliphatic glucosinolates are different among the three mustards, but the indolic glucosinolate kinds are exactly the same among the three mustards. Why?
- 4) The English writing should be improved, and some mistakes exist in the manuscript, please check and revise them carefully.
 Line 43-45 on Page 1, Line 18-20 on Page 3, the sentences are difficult to understand, please rewrite them.
 Line 25 on Page 3, delete 'a'
 Line 44 on Page 4, change 'obsereved' to 'observed'
 Line 37 on Page 8, change 'not only pickles' to 'is not only pickled'

Review form: Reviewer 2**Is the manuscript scientifically sound in its present form?**

Yes

Are the interpretations and conclusions justified by the results?

Yes

Is the language acceptable?

Yes

Is it clear how to access all supporting data?

Not Applicable

Do you have any ethical concerns with this paper?

No

Have you any concerns about statistical analyses in this paper?

No

Recommendation?

Accept with minor revision (please list in comments)

Comments to the Author(s)

This is overall a good study and manuscript. However I do have a few comments and suggestions. Please find the attached (Appendix A) and revise accordingly.

Thank you.

Decision letter (RSOS-182054.R0)

03-Jan-2019

Dear Dr Tang:

Title: Variations in the glucosinolates of the individual edible parts of three stem mustards
Manuscript ID: RSOS-182054

Thank you for submitting the above manuscript to Royal Society Open Science. On behalf of the Editors and the Royal Society of Chemistry, I am pleased to inform you that your manuscript will be accepted for publication in Royal Society Open Science subject to minor revision in accordance with the referee suggestions. Please find the reviewers' comments at the end of this email.

The reviewers and handling editors have recommended publication, but also suggest some minor revisions to your manuscript. Therefore, I invite you to respond to the comments and revise your manuscript.

Because the schedule for publication is very tight, it is a condition of publication that you submit the revised version of your manuscript before 12-Jan-2019. Please note that the revision deadline will expire at 00.00am on this date. If you do not think you will be able to meet this date please let me know immediately.

When submitting your revised manuscript, you will be able to respond to the comments made by the referees and upload a file "Response to Referees" in "Section 6 - File Upload". You can use this

to document any changes you make to the original manuscript. In order to expedite the processing of the revised manuscript, please be as specific as possible in your response to the referees.

Best wishes,
Dr Laura Smith
Publishing Editor, Journals

RSC Associate Editor:
Comments to the Author:
(There are no comments.)

RSC Subject Editor:

Comments to the Author:
(There are no comments.)

Reviewer comments to Author:
Reviewer: 1

Comments to the Author(s)
Comments to the Author(s)

This is an acceptable paper for publication in Royal Society Open Science. The manuscript entitled " Variations in the glucosinolates of the individual edible parts of three stem mustards " is an interesting and nice work. The manuscript reported the composition and content of glucosinolates among the edible parts (petioles, peel, and flesh) of tuber mustard, bamboo shoots mustard, and baby mustard. The results also well explained the relationships among the preparation methods and the glucosinolate profiles of the three stem mustards, and this work will provide a reference for the human daily diet.

However, some comments should be made to improve the manuscript. Below are some questions and/or suggestions.

- 1) Line 18-19 on Page 2, as I know it is more than 200 kinds, please check it.
- 2) Line 30-43 on Page 7, what are the genetic relationships between the three mustards in this manuscript and the species you are comparing? What caused different glucosinolate components among them?
- 3) Line 19-32 on Page 8, this is a very interesting phenomenon. The kinds of aliphatic glucosinolates are different among the three mustards, but the indolic glucosinolate kinds are exactly the same among the three mustards. Why?
- 4) The English writing should be improved, and some mistakes exist in the manuscript, please check and revise them carefully.
Line 43-45 on Page 1, Line 18-20 on Page 3, the sentences are difficult to understand, please rewrite them.
Line 25 on Page 3, delete 'a'
Line 44 on Page 4, change 'obsereved' to 'observed'
Line 37 on Page 8, change 'not only pickles' to 'is not only pickled'

Reviewer: 2

Comments to the Author(s)

This is overall a good study and manuscript. However I do have a few comments and suggestions. Please find the attached and revise accordingly.
Thank you.

Author's Response to Decision Letter for (RSOS-182054.R0)

See Appendix B.

Decision letter (RSOS-182054.R1)

14-Jan-2019

Dear Dr Tang:

Title: Variations in the glucosinolates of the individual edible parts of three stem mustards
(*Brassica juncea*)
Manuscript ID: RSOS-182054.R1

It is a pleasure to accept your manuscript in its current form for publication in Royal Society Open Science. The chemistry content of Royal Society Open Science is published in collaboration with the Royal Society of Chemistry.

RSC Associate Editor
Comments to the Author:
(There are no comments.)

Reviewer(s)' Comments to Author:

Appendix A

Variations in the glucosinolates of the individual edible parts of three stem mustards

Abstract

- Page 1, Lines 48-56: The flow is not smooth. The key is sinigrin was the predominant glucosinolate in all mustard investigated. I would suggest to show the relative amount of sinigrin in each mustard rather than indole glucosinolate to show the most important information and make the abstract little less lengthy.

Introduction

- Page 2, Line 20: Brassica is a more proper term instead of crucifers these days.
- Page 2, Line 27: I would suggest to change “unique location” to “different cell, physically separated from its substrate glucosinolates”.
- Page 2, Line 52-53: The hydrolysis products of glucosinolates contribute to the pungent flavor, not glucosinolate itself.

Experimental

- Page 3, Line 13-14: ‘cultivar’ not ‘variety’. Change it throughout the manuscript.
- Page 3, Line 21: What do you mean by ‘similar growing tendency’? Similar size? Please clarify.
- Page 3, Line 23: What is your definition of “peel” and “flesh”? Glucosinolates can be hydrolyzed during cutting and/or peeling process. Please provide more details on this step.
- Page 3, sample prep and HPLC condition: At which step was internal standard added? What was the mobile phase condition? Was it 100% water and 100% acetonitrile? Was isocratic system used or gradient system? How did you identify and quantify glucosinolates? Provide more details.

Results

3.1. Glucosinolate composition in the mustard

- Line 31: Italicize “Brassica”
- This is a section about composition. Why don't you provide % of important glucosinolates? Maybe aliphatic, indole and glucosinolates in each species? Or % of one or two most abundant

glucosinolates would also be helpful. It is mentioned later in discussion but this information should also be in result.

3.2. Total glucosinolates

- Line 42: Delete “respectively”. This is not needed.

3.4. Indole glucosinolates

Line 29: Change “obviously” to “significantly”.

3.6. Principal component analysis

- PCA and PLS-DA are different. Therefore, it seems little odd to see the PLS-DA in PCA section. You need to change the title of this section.

- What compounds were most distinguishable by PLS-DA? You can use the VIP value and select a biomarker that were different among species or among tissues. Fig. 6C itself is not readable that much and therefore you need to read in the text.

Discussion

- Page 7, Line 59-: “however, it is gluconapin in Chinese kale [18] and most varieties of *B. rapa*. The predominant glucosinolate in a few other varieties of *B. rapa* is glucobrassicinapin [24]” is not needed. These are different crop, different species from your materials and therefore, it is meaningless to directly compare like this. If you still want to cite these references (18 and 24), re-phrase so that it flows better.

- Page 8, Line 3: Change “cruciferous” to “*Brassica*”. Change it throughout the manuscript if there are more.

- Page 8, Line 31: “close” not “closely”

- Page 8, Lines 46-52: On what ground do you say that tuber mustard requires the most processing, baby mustard requires less processing, and bamboo shoots mustard should be eaten fresh? Is this because of sinigrin content? If so, this is irony because you say that AITC is effective against cancer and suggest to consume mustard daily.

Also, you need to carefully check references. Are ITCs anti-cancer or anti-carcinogenic? Cancer and carcinogenesis are different process. Please don't be confused or use the terms interchangeably.

Page 8, Lines 54-58: What is the ground of sugar masking pungent flavor? The total glucosinolate content of 3 mustards is not compared in one figure. You need to show a stacked bar graph showing aliphatic, indole and aromatic glucosinolates (one bar) in 3 mustards. And then you need to discuss what causes more pungent flavor of tuber mustard than baby mustard although sinigrin content is not different.

If you want to refer [13], higher amount of myrosinase in tuber mustard than in baby mustard, you can also check the myrosinase activity. Check Kim et al. (Cultivar-specific changes in primary and secondary metabolites in pak choi (*Brassica rapa*, *Chinensis* group) by methyl

jasmonate, 2017. Int. J. Mol. Sci.) and Frazie et al. (Health-Promoting Phytochemicals from 11 Mustard Cultivars at Baby Leaf and Mature Stages. 2017. Molecules) for more information.

Figure 5: Delete the total aromatic glucosinolate figure. Since gluconasturtiin is the only aromatic glucosinolate detected in this study, the total content figure is redundant.

Others

- Mustard can be either *Brassica juncea* or *B. nigra*. *B. nigra* is never mentioned in this article. And if you keep using “mustard” without clarifying the scientific name, it can confuse or mislead. I would suggest to put “*Brassica juncea*” in the article title so that the readers know this article only focuses on *B. juncea*.
- In abstract, Page 3 Line 5, and Page 9 Line 29, what do you mean by “preparation method”? This is unclear.
- It is said that this article provides a reference for the human daily diet. But it doesn't seem true. Although how to consume each mustard is suggested, it cannot be seen as “a reference for the human daily diet” especially on what ground you suggest that way is not clear.
- Throughout the manuscript, why you separated tissues is not shown. Therefore, the importance of the information in this manuscript is not strong enough. Please include this information to justify this study.

Appendix B

Manuscript ID: RSOS-182054

Title: Variations in the glucosinolates of the individual edible parts of three stem mustards

Dear Professor Anthony Stace FRS

We are pleased to submit a revised version of the manuscript RSOS-182054. We are truly grateful to reviewers' critical comments and thoughtful suggestions. We have worked through the details in the comments of the reviewers, and revised the paper accordingly. We hope the new manuscript will meet the standard of *Royal Society Open Science*. Below you will find our point-by-point responses to the reviewers' comments/questions.

Thank you very much for your consideration.

Yours sincerely,

Hao-Ru Tang

Referees' Comments to Author:

Referee: 1

Comments to the Author(s)

1) Line 18-19 on Page 2, as I know it is more than 200 kinds, please check it.

Thanks for your suggestion. We have revised it in the manuscript.

2) Line 30-43 on Page 7, what are the genetic relationships between the three mustards in this manuscript and the species you are comparing? What caused different glucosinolate components among them?

It is a very good suggestion. All of the species compared in our manuscript are belong to Brassica. Cauliflower, cabbage, and Chinese kale are three different variants of *Brassica oleracea*. Korean leaf mustard (*B. juncea* var. *integrifolia*) and the three stem mustard in our manuscript are different variants of *B. juncea*. Glucosinolate is one of the most important secondary metabolites in Brassica vegetables. However, the genes in different Brassica vegetables changed after a long period of evolution, and the biosynthesis of glucosinolates was affected by the evolution, which caused the difference in glucosinolate components.

3) Line 19-32 on Page 8, this is a very interesting phenomenon. The

kinds of aliphatic glucosinolates are different among the three mustards, but the indolic glucosinolate kinds are exactly the same among the three mustards. Why?

It is a very good suggestion. The aliphatic glucosinolates generally related with flavor, and the indole glucosinolates generally contribute to the plant defense against pathogens and generalist herbivores. Besides, aliphatic glucosinolates controlled by genetic factors, and indole glucosinolates major influenced by environment. Different vegetables are different in flavor, however indole glucosinolates maybe indispensable in plants. Therefore, the composition of aliphatic glucosinolates in the three stem mustard varied greatly, and the composition of indole glucosinolates is similar.

4) The English writing should be improved, and some mistakes exist in the manuscript, please check and revise them carefully.

Line 43-45 on Page 1, Line 18-20 on Page 3, the sentences are difficult to understand, please rewrite them.

Line 25 on Page 3, delete 'a'

Line 44 on Page 4, change 'obsereved' to 'observed'

Line 37 on Page 8, change 'not only pickles' to 'is not only pickled'

Thanks for your suggestion. We have carefully checked and revised the manuscript to improve the English expression by ourselves, and then the manuscript was also critically read by Dr. Yunting Zhang (University of California, Davis).

Referee: 2

Comments to the Author(s)

Abstract

1). - Page 1, Lines 48-56: The flow is not smooth. The key is sinigrin was the predominant glucosinolate in all mustard investigated. I would suggest to show the relative amount of sinigrin in each mustard rather than indole glucosinolate to show the most important information and make the abstract little less lengthy.

Thanks for your suggestion. We supplemented the relative content of sinigrin in results, and the content about indole glucosinolates was removed from the abstract according to the recommendations.

Introduction

1) - Page 2, Line 20: Brassica is a more proper term instead of crucifers these days.

Thanks for your suggestion. We have revised them in the manuscript.

2) - Page 2, Line 27: I would suggest to change “unique location” to “different cell, physically separated from its substrate glucosinolates”.

Thanks for your suggestion. We have revised it in the manuscript.

3) - Page 2, Line 52-53: The hydrolysis products of glucosinolates contribute to the pungent flavor, not glucosinolate itself.

Thanks for your suggestion. We have revised it in the manuscript.

Experimental

1) - Page 3, Line 13-14: ‘cultivar’ not ‘variety’. Change it throughout the manuscript.

Thanks for your suggestion. We have revised them in the manuscript.

2) - Page 3, Line 21: What do you mean by ‘similar growing tendency’? Similar size? Please clarify.

Thanks for your suggestion. Our previous description is inaccurate, and our meaning is the similar size. The details have been modified in the manuscript.

3) - Page 3, Line 23: What is your definition of “peel” and “flesh”? Glucosinolates can be hydrolyzed during cutting and/or peeling process. Please provide more details on this step.

Thanks for your suggestion. All of the three mustards have a clear dividing line between peel and flesh. We sampled according to the dividing line. Glucosinolates can be hydrolyzed easily just as you said. After sampling, we quickly froze the samples in liquid nitrogen, then store them at -80 °C, and finally lyophilized them. The details have been added to the manuscript.

4) - Page 3, sample prep and HPLC condition: At which step was internal standard added? What was the mobile phase condition? Was it 100% water and 100% acetonitrile? Was isocratic system used or gradient system? How did you identify and quantify glucosinolates? Provide more details.

Thanks for your suggestion. The internal standard added to supernatant after centrifugation. The mobile phase was 100% water and 100% acetonitrile, and the gradient system was used in our experiment. The glucosinolates were identified by liquid chromatography-mass spectrometry (LC-MS), and quantified by a standard curve. More details have been added to the manuscript.

Results

3.1. Glucosinolate composition in the mustard

1) - Line 31: Italicize “Brassica”

Thanks for your suggestion. We have revised it in the manuscript.

2) - This is a section about composition. Why don't you provide % of important glucosinolates? Maybe aliphatic, indole and glucosinolates in each species? Or % of one or two most abundant glucosinolates would also be helpful. It is mentioned later in discussion but this information should also be in result.

Thanks for your suggestion. The proportions of sinigrin have been added to the manuscript. .

3.2. Total glucosinolates

1) - Line 42: Delete “respectively”. This is not needed.

Thanks for your suggestion. We have revised it in the manuscript.

3.4. Indole glucosinolates

1) - Line 29: Change “obviously” to “significantly”.

Thanks for your suggestion. We have revised it in the manuscript.

3.6. Principal component analysis

1) - PCA and PLS-DA are different. Therefore, it seems little odd to see the PLS-DA in PCA section. You need to change the title of this section.

Thanks for your suggestion. We have revised it in the manuscript.

2) - What compounds were most distinguishable by PLS-DA? You can use the VIP value and select a biomarker that were different among species or among tissues. Fig. 6C itself is not readable that much and therefore you need to read in the text.

Thanks for your suggestion. We have revised it in the manuscript, and Fig 6C has been explained in the title of Fig. 6.

Discussion

1) - Page 7, Line 59-: “however, it is gluconapin in Chinese kale [18] and most varieties of *B. rapa*. The predominant glucosinolate in a few other varieties of *B. rapa* is glucobrassicinapin [24]” is not needed. These are different crop, different species from your materials and

therefore, it is meaningless to directly compare like this. If you still want to cite these references (18 and 24), re-phrase so that it flows better.

Thanks for your suggestion. We have revised it in the discussion.

2) - Page 8, Line 3: Change “cruciferous” to “Brassica”. Change it throughout the manuscript if there are more.

Thanks for your suggestion. We have revised them in the manuscript.

3) - Page 8, Line 31: “close” not “closely”

Thanks for your suggestion. We have revised it in the manuscript.

4) - Page 8, Lines 46-52: On what ground do you say that tuber mustard requires the most processing, baby mustard requires less processing, and bamboo shoots mustard should be eaten fresh? Is this because of sinigrin content? If so, this is irony because you say that AITC is effective against cancer and suggest to consume mustard daily.

Thanks for your suggestion. A high content of sinigrin is effective against cancer, but it tastes bad. Taste is the most important in the consumption of vegetables. Besides, the custom of processing already exists in life. We just explained the relationship between the content of glucosinolates and the custom of processing.

5) Also, you need to carefully check references. Are ITCs anti-cancer or anti-carcinogenic? Cancer and carcinogenesis are different process.

Please don't be confused or use the terms inter-changeably.

Thanks for your suggestion. We have revised it in the manuscript.

6) Page 8, Lines 54-58: What is the ground of sugar masking pungent flavor? The total glucosinolate content of 3 mustards is not compared in one figure. You need to show a stacked bar graph showing aliphatic, indole and aromatic glucosinolates (one bar) in 3 mustards. And then you need to discuss what causes more pungent flavor of tuber mustard than baby mustard although sinigrin content is not different. If you want to refer [13], higher amount of myrosinase in tuber mustard than in baby mustard, you can also check the myrosinase activity. Check Kim et al. (Cultivar-specific changes in primary and secondary metabolites in pak choi (*Brassica rapa*, *Chinensis* group) by methyl jasmonate, 2017. *Int. J. Mol. Sci.*) and Frazie et al. (Health-Promoting Phytochemicals from 11 Mustard Cultivars at Baby Leaf and Mature Stages. 2017. *Molecules*) for more information.

① **Thanks for your suggestion. The presumption is based on life experience.**

② **It is a good suggestion. We have added the stacked bar graph in Supplemental Figure 1 as follow.**

Supplemental Fig. 1 The concentration and proportion of individual glucosinolates in different tissues of the three stem mustards. A: glucosinolate concentration. B: proportion.

③ The papers suggested in your comment are very important. We have added them to the references after carefully reading.

7) Figure 5: Delete the total aromatic glucosinolate figure. Since

gluconasturtiin is the only aromatic glucosinolate detected in this study, the total content figure is redundant.

Thanks for your suggestion. We have revised it in the manuscript.

Others

1) - Mustard can be either Brassica juncea or B. nigra. B. nigra is never mentioned in this article. And if you keep using “mustard” without clarifying the scientific name, it can confuse or mislead. I would suggest to put “Brassica juncea” in the article title so that the readers know this article only focuses on B. juncea.

It is an excellent suggestion. We have revised it in the manuscript.

2) - In abstract, Page 3 Line 5, and Page 9 Line 29, what do you mean by “preparation method”? This is unclear.

Thanks for your suggestion. In order to express it more clearly, the “preparation” has been changed to “cooking”.

3) - It is said that this article provides a reference for the human daily diet. But it doesn't seem true. Although how to consume each mustard is suggested, it cannot be seen as “a reference for the human daily diet” especially on what ground you suggest that way is not clear.

Thanks for your suggestion. We have revised it in the manuscript.

4) - Throughout the manuscript, why you separated tissues is not shown. Therefore, the importance of the information in this manuscript is not strong enough. Please include this information to justify this

study.

Thanks for your suggestion. We have added the reason in the introduction.